# Computational Models for Clinical Applications in Personalized Medicine—Guidelines and Recommendations for Data Integration and Model Validation

**DOI:** 10.3390/jpm12020166

**Published:** 2022-01-26

**Authors:** Catherine Bjerre Collin, Tom Gebhardt, Martin Golebiewski, Tugce Karaderi, Maximilian Hillemanns, Faiz Muhammad Khan, Ali Salehzadeh-Yazdi, Marc Kirschner, Sylvia Krobitsch, Lars Kuepfer

**Affiliations:** 1Novo Nordisk Foundation Center for Protein Research, Faculty of Health and Medical Sciences, University of Copenhagen, 2200 N Copenhagen, Denmark; catherine.bjerre.collin@cpr.ku.dk (C.B.C.); tugce.karaderi@sund.ku.dk (T.K.); 2Department of Systems Biology and Bioinformatics, University of Rostock, 18057 Rostock, Germany; tom.gebhardt@uni-rostock.de (T.G.); maximilian.hillemanns@uni-rostock.de (M.H.); faiz.khan3@uni-rostock.de (F.M.K.); 3Heidelberg Institute for Theoretical Studies gGmbH, 69118 Heidelberg, Germany; martin.golebiewski@h-its.org; 4Center for Health Data Science, Faculty of Health and Medical Sciences, University of Copenhagen, 2200 N Copenhagen, Denmark; 5Max-Planck-Institute for Multidisciplinary Sciences, 37077 Göttingen, Germany; ali.salehzadeh-yazdi@mpinat.mpg.de; 6Forschungszentrum Jülich GmbH, Project Management Jülich, 52425 Jülich, Germany; m.kirschner@fz-juelich.de (M.K.); s.krobitsch@fz-juelich.de (S.K.); 7Institute for Systems Medicine with Focus on Organ Interaction, University Hospital RWTH Aachen, 52074 Aachen, Germany

**Keywords:** personalized medicine, computational models, data integration, model validation, guidelines and recommendations, clinical translation, ethical and legal requirements

## Abstract

The future development of personalized medicine depends on a vast exchange of data from different sources, as well as harmonized integrative analysis of large-scale clinical health and sample data. Computational-modelling approaches play a key role in the analysis of the underlying molecular processes and pathways that characterize human biology, but they also lead to a more profound understanding of the mechanisms and factors that drive diseases; hence, they allow personalized treatment strategies that are guided by central clinical questions. However, despite the growing popularity of computational-modelling approaches in different stakeholder communities, there are still many hurdles to overcome for their clinical routine implementation in the future. Especially the integration of heterogeneous data from multiple sources and types are challenging tasks that require clear guidelines that also have to comply with high ethical and legal standards. Here, we discuss the most relevant computational models for personalized medicine in detail that can be considered as best-practice guidelines for application in clinical care. We define specific challenges and provide applicable guidelines and recommendations for study design, data acquisition, and operation as well as for model validation and clinical translation and other research areas.

## 1. Introduction

The amount of personalized data in today’s medicine is continuously increasing and holds great promises for both diagnosis and therapy at the single patient level. In the face of these complex and heterogenous data volumes, computational models support a functional understanding of the mechanisms and factors that drive certain diseases. Likewise, they allow the design of personalized treatment strategies in response to central clinical questions [1]. Computational models thus have the potential to translate in vitro, preclinical and clinical results (and their related uncertainty) into descriptive or predictive expressions. Over the last decades, the added value of such models, also called digital evidence, in medicine and pharmacology has increasingly been recognized by the scientific community [2,3], as well as regulatory bodies, including the US Food and Drug Administration (FDA) or the European Medicines Agency (EMA) [4,5,6,7]—irrespective of their ultimate use or application. Computational models are now integrated in different fields in medicine and drug development expanding from disease modelling and biomarker research to the assessment of drug efficacy and safety. In silico processing and interpretation of clinical measurements can be either data-driven or theory-based (Figure 1). Though, both concepts are rather complementary, and they share the common general requirements for data standardization and data documentation. Mechanistic models aim for a structural representation of the governing physiological processes in the model equations to support a functional understanding of the underlying mechanisms. Vice versa, data-driven approaches, such as machine learning (ML) and deep learning (DL), which use algorithms and models that try to mimic (human) intelligence and that are commonly referred to as artificial intelligence (AI) [8,9], aim for knowledge discovery in big data through multidimensional regression analysis. In consequence, mechanistic models require a structural understanding of a process, yet the demand for data can be limited. Concepts in machine learning, in turn, are fundamentally based on large data sets, yet these models do not necessarily need any prior functional understanding (Figure 1). Applied in personalized medicine, these modelling approaches allow stratification of patients into specific groups with similar characteristics—a prerequisite for advanced diagnosis, targeted therapies, and prevention strategies. In the following section, the most relevant modelling approaches for clinical applications in personalized medicine will be briefly introduced.

## 2. Modelling Approaches for Clinical Applications in Personalized Medicine

### 2.1. Mechanistic Models

The aim of a mechanistic model is to functionally understand, examine, and predict the emergent properties of individual components of a biological system and the manner in which they are coupled. It also predicts the complex non-linear dynamics of system variables and simulates the underlying dysregulations in processes that drive a healthy state into a specific disease. To deal with this complexity, various systems biology and systems medicine approaches have been developed. By incorporating biochemical, physiological, and environment interactions, these approaches produced successful results to understand the non-intuitive behavior of biological systems and especially of the human body [10,11,12]. Models are developed from available knowledge describing physical/bio-chemical relationships among species, and kinetic parameters are retrieved from data under investigation to mimic the biological reality. After calibration, simulations are performed that generate new hypotheses to design new experiments. If experiments validate model-based hypotheses, it will generate new knowledge, which may be used for constructing another model, thus iteratively improving the functional understanding. In this way, the systems-biology/systems-medicine approach iterates in data-driven models and model-driven experimentations. Models are an abstract representation of reality, and thereby their validity and usefulness depends on the context and assumptions being made. Modeling approaches have different levels of abstraction, predictive power, advantages, and limitations. Previously established concepts range from static molecular interaction maps and constraint based modelling to qualitative logic-based models to more detailed quantitative kinetic models. The choice of a model formalism depends on the availability of data, the type of research question and the size and structure of the system. The most relevant mechanistic models are described in the following section.

#### 2.1.1. Molecular Interaction Maps

Molecular interaction maps (MIMs) are static models that depict the physical and causal interactions among biological species in the form of networks [13]. They serve as a knowledge-base containing information about different pathways and regulatory modules involved in a disease, such as Parkinson’s [14] or signaling in cancer [15]. MIMs can be computationally analyzed using graph-theory concepts to identify network static properties such as (i) identification of the most influential nodes, (ii) community detection by a clustering approach, and (iii) link prediction for the discovery of hidden links. Furthermore, upon overlying expression data, such maps serve as visualization tools for the activity level of regulators and their targets of established disease markers, which will provide the simplest mechanistic visualization of data.

#### 2.1.2. Constraint-Based Models

Constraint-based models, such as GEnome-scale Metabolic models (GEM), provide a mathematical framework to gaining an understanding of metabolic capacities of a cell, enabling system-wide analysis of genetic perturbations, exploring metabolic diseases, and finding the essential enzymatic reactions as well as drug targets [16]. GEMs have received substantial attention, and many investigations have been done about their applications in different aspects of medical sciences. This modeling approach has been applied to various fields, ranging from cancer [17] to obesity [18] and Alzheimer disease [19].

#### 2.1.3. Boolean Models

Boolean modelling (BM) is the simplest form of logic-based models where nodes (e.g., a gene, protein, a transcription factor, or microRNA, etc.) are described by one of two possible states: 1 (ON, activation) or 0 (OFF, inactivation) [20]. The regulatory relationship from upstream nodes (regulators) to downstream nodes (targets) are expressed by the logical operators AND, OR, and NOT. Such models do not require detailed kinetic data for parameter estimation, which makes them fit to apply to large biological systems and easier/quicker to calibrate/train with data. In the context of systems medicine, this approach is often applied for cancer research [21,22].

#### 2.1.4. Quantitative Models

Quantitative modeling, such as the ordinary differential equations (ODEs)-based approach, quantitatively analyses the behavior of a biochemical reaction over time. It comprises a set of differential equations containing variables (which are quantities of interest, e.g., the concentration of biological species) and parameters that describe how the system responds to different stimuli or perturbations (an extensive resource of ODE models represents the BioModels database [23]). Parameter values are fixed and can be personalized by either each experimental setup or a patient to make a model match the given data. This approach describes biological-systems dynamics in detail; however, it usually applies for a single pathway or only a few reactions due to the requirement of detailed kinetic data for parameter estimations. In personalized medicine in the context of patient stratification, ODE models are applied for individual biomarker discovery [24], drug response, and tailored treatments [25].

#### 2.1.5. Pharmacokinetic Models

Pharmacokinetic models are a particular application of ODE models that describe the concentration of a drug in plasma or different tissues. Drug pharmacokinetics are frequently used as a surrogate for drug-induced responses since they may be used to estimate on-target and off-targeted drug exposure and in turn a to-be-expected effect strength. They can either be described by compartmental pharmacokinetic (PK) modelling [26] or by physiologically based PK (PBPK) modeling [27,28,29]. Compartmental PK models, also referred to as population PK (popPK) models, are top-down models that derive an empirical model structure from plasma PK. Model building usually starts with a simple one-compartment model, which is extended by linear absorption as well as clearance rates. Changes in model structure such as peripheral compartments may become necessary during model development. There are domain-specific standard formats available to structure and share them, such as PharmML (Pharmacometrics Markup Language [30]) as an exchange format for the encoding of models, associated tasks, and their annotation in pharmacometrics. In contrast to compartmental PK approaches, PBPK modelling aims to reproduce the physiology of an organism at a large level of detail [29,31,32]. Different organs are explicitly represented in a PBPK model, and they are assigned specific physiological properties such as volumes, surface, composition, and blood-flow rates. PBPK modelling allows to integrate very diverse patient-specific information ranging from the molecular scale to physiological properties at the whole-body level. This is because of the large granularity of PBPK models, which may represent physiological information from different levels of biological organization. A frequent application for the individualization of PBPK models is their specification to represent specific cohorts of patients such as elderly [33] or diseased patients from a base reference model [34]. This benchmark usually represents an average individual with mean values of physiological parameters.

#### 2.1.6. Software Resources and Tools

In the following, we provide a list of widely used resources and tools for the construction, visualization, and simulation of MIMs, including qualitative and quantitative models and pharmacokinetic models (summarized in Table 1). The choice of a single resource or tool is difficult; however, some resources, e.g., OmniPath [35] and Regulatory INteraction Graph (RING) [36], retrieve interactions from multiple repositories, which can, for example, ease MIM construction. Similarly, each tool is designed to address specific challenges in modeling, e.g., CellNetAnalyzer (CNA) [37] can be used to easily simulate a model for large combinations of inputs and perturbations (tens of thousand), but encoding models with graphical view is time consuming. CellCollective [38] is a web-based simulation tool for community-driven model development and requires no complex mathematical equations to encode. Gene Interaction Network simulation suite (GINsim) [39] is a powerful tool for attractor analysis but for a large model and a large number of inputs and perturbations, it may not be ideal.

Several tools and resources can be combined to workflows for model development, simulation, and publication. For example, a model could be constructed in CellDesigner [63], simulated in JWS Online [60] and published in BioModels [47]. Community-driven harmonization efforts led to several standards and formats that support the interoperability of the tools and resources. Over time, different, partially overlapping communities have been established, which take responsibility for the maintenance and development of the standards and formats. The Computational Modeling in Biology Network (COMBINE) community [64] develops standards to store and exchange computational models. Prominent examples are the Systems Biology Markup Language (SBML) [65], CellML [66] and The Systems Biology Graphical Notation (SBGN) [67]. Their standards are supported by many tools for MIMs and quantitative models. The Disease Maps Project [68] is a large-scale community with the goal to enhance the understanding of diseases. Their efforts focus on MIMs, including map and tool development. The consortium for Boolean models is CoLoMoTo (http://colomoto.org), which develops standards for model representation and exchange, especially SBML Qual [69].

### 2.2. Machine Learning and Deep Learning

Data-driven approaches treat the causal mechanism as unknown and aim to model a function that operates on large-scale data input to predict the outcome, regardless of the unknown physiological processes. Since the mechanisms operating in the systems being modelled, i.e., which factors together drive outcomes, are considered too complex to be determined, ML and DL are often referred to as black-box models. Consequently, the rationale for the outputs generated is inscrutable not only by physicians but also by the engineers who develop them. There are ambitious attempts to make data-driven models explainable or understandable [8] by providing information on underlying model mechanisms and the factors driving predictions. In the context of personalized medicine, this requires examination of the concepts of explainee, explanandum, and explanans, amongst others. For example, the explainee, the person to whom an explanation is addressed, e.g., a patient, clinician, or researcher, will determine what constitutes an appropriate explanation of model predictions [70]. The object of the explanation (the explanandum) is not the patient/the disease course but the model/the model’s prediction, and the explanans that explains the prediction may in itself be complex mathematics. Clear causality is not provided, although explanations may make intuitive sense to explainees.

For example, models that provide post-hoc feature-importance visualisation often show age or pre-existing disease to be features that drive predictions toward a negative outcome. While this does not give information about causality, the explainee “provides” the causal link in his/her own interpretation of the explanation. Clear causal understanding is not provided, although data-driven models can be hypothesis-generating and can therefore provide clues to understanding. Understanding is epistemologically complex, but an explainee (for example, a biologist or a clinician) can be said to understand when they are in a position to move forward. For example, a post-hoc model providing visualization of the factors driving a clinical prediction for a patient can give a clinical sufficient understanding to move forward with recommending a clinical course of action to her patient [71]. A biologist can be said to understand when a biological area has been identified which seems to recur in model-prediction feature drivers; or a bioinformatician can be said to understand when reapplying the model’s code and examining output.

The quality of such black-box models is assessed through the accuracy of their predictions, which are tested in a variety of ways. As a subset of AI, ML works cyclically and learns through experience and is largely used for prediction models and pattern recognition (Figure 1). A special case of ML that is used for integration of complex data such as omics and clinical data is DL. In DL, Deep Neural Networks can extract and process information from given data. DL mimics the human brain via connecting multiple artificial neurons in deep and densely connected layers [72,73,74]. ML/DL approaches provide an opportunity to move away from interpretive attempts to apply group-level associations and instead predict responses in individual patients, i.e., enabling more personalized medicine. Group-level associations, unlike individual-level predictions, are interpreted in a largely unstructured way for the treatment of an individual and may not provide the optimal treatment approach for that individual. Thus, classical group-based clinical studies and ML/DL approaches represent two distinct paradigms: the first provides information based on associations at the level of a patient group, while the second, the ML/DL approach, builds a prediction at the level of an individual patient [75,76].

ML attempts ever more accurate clustering or classification to a high level of accuracy and a high level of confidence. Different types of ML are employed as appropriate according to input data and the aim of analysis, i.e., according to the clinical question. ML approaches learn the theory automatically from the data through a process of inference, model fitting, or learning from examples [77]. ML can be supervised, unsupervised, or semi-supervised. Unsupervised learning has the potential to take all features into account, to comprise dimensionality reduction, permit feature elicitation, and big-data visualization, all of which allow for better understanding of big medical data and the factors driving disease initiation and progression. Unsupervised learning offers the ability to discover new knowledge, which can be used to refine outcome prediction for the individual patient. In supervised learning, the model is supplied with labeled input features that are considered when predicting a predetermined outcome from new data, either through regression (for continuous results such as number of days or months before disease debut) or classification (for discrete results such as survival/death or for image classification). Supervised ML clinical-decision support tools allow ML the possibility to tailor prediction to the individual patient. Semi-supervised learning employs both labelled and unlabeled data for training. In all of these ML approaches, model validation is essential, and the accuracy of ML results is verified using independent test sets [74].

Even though best practices of ML have been developed, they are either not applied at all or applied only partially for selected aspects of the models, meaning an inconsistent or incomplete documentation and reporting as well as sharing of ML models and their evaluation [78]. This issue often results in vague and inconsistent decision making by the ML algorithms when re-used, making it difficult or even impossible to reproduce the results, with many uncertainties [79]. Particularly in clinical research and application, it is absolutely mandatory to develop and report ML/DL-derived models following established standards for gaining trust in the resulting decision making. In this context, especially reporting guidelines for the corresponding models and their validation are most relevant providing a list of attributes and elements to describe them properly. Recent examples are the SPIRIT-AI [80] and CONSORT-AI [81] checklists for the reporting of clinical trials that involve ML methods. The AIMe registry for AI in biomedical research (https://aime-registry.org) recently has been introduced as a community-driven platform for reporting biomedical AI systems [82]. The AIMe registry has a web interface that guides authors of new ML algorithms through the newly developed AIMe standard, a generic minimal information standard that allows the reporting of any biomedical AI system and is divided into five sections: Metadata, purpose, data, method, and reproducibility.

## 3. Models in Clinical Research for Discovery, Diagnosis, and Therapy

It is to be expected that patients will greatly benefit in the future from developments that equip personalized medicine with predictive capabilities to investigate in silico clinically relevant questions. Currently, there are a number of computational-modelling approaches in pre-clinical and clinical research that are able to address these questions in greater detail and, therefore, play a leading role for the future development of personalized medicine. The following section contains an illustrative overview of successful computational analyses in the discovery, diagnosis, and therapy of clinical research.

### 3.1. Discovery

Model applications in discovery are usually mechanism-based, such as MIMs, GEMs, BMs, and ODEs, since they are frequently hypothesis driven. This is because the availability of data at this level is commonly not sufficient for purely data-driven analyses. Mechanistic models in discovery play a significant role in a wide range of clinically relevant questions ranging from representation of disease mechanisms to identification of drug targets or simulations of disease-specific phenotypes (summarized in Table 2):

A recently published map on inflammation resolution provides functionality to visualize Omics data and allows making hypotheses on the role of connected molecules in a disease phenotype [83]. Another well-known example for MIMs is the disease map of Parkinson’s [14]. These maps serve as a knowledge platform and represent the mechanisms of the disease in a standardized visualization. Thereby, they structure the growing knowledge of the field in a comprehensible manner. Another interesting example of MIM is the atlas of the cancer signaling network [15], which depicts in detail the molecular mechanisms involved in cancer. High-throughput data can be visualized on the map to perform functional analysis and identify dysregulated pathways. Wu et al. constructed a comprehensive molecular interaction map for rheumatoid arthritis containing detailed molecular mechanisms of the processes in patients affected by rheumatoid arthritis [84]. The map was analyzed for topological properties to suggest diagnostic and therapeutic markers for rheumatoid arthritis.

Disease-specific GEMs were used for the identification of biomarkers and drug targets in metabolism-related disorders including: cancer [17], type 2 diabetes [87], obesity [18], non-alcoholic fatty liver disease (NAFLD) [88], and Alzheimer’s disease (AD) [19]. In 2017, Uhlen et al. generated GEMs of 17 types of cancer by integrating transcriptome data into a network of human metabolism using the task-driven integrative network inference for tissues (tINIT) method [91]. In addition to predicting driver genes for tumor growth, they demonstrated a widespread metabolic heterogeneity in different patients, highlighting the necessity of personalized medicine for cancer treatments [17]. Varemo et al. generated manually curated GEMs to identify signatures of a diabetic muscle. They suggested a gene signature, which successfully classified the disease progression of individual samples [87]. Mardinoglu et al. (2013) reconstructed an adipocyte-specific GEM and showed that the metabolic activity of androsterone and ganglioside GM2 increased in obese subjects, and their mitochondrial metabolic activities decreased compared to lean subjects [18]. They also identified chondroitin and heparan sulphates as suitable biomarkers for the staging of NAFLD by analyzing the reconstructed iHepatocytes2322 [88]. In 2014, Stempler et al. reconstructed an AD-specific GEM and predicted several metabolic biomarkers of the AD progression, including succinate and prostaglandin D2 [19].

Patient-specific data have been integrated in Boolean models amongst others to simulate patient disease phenotypes and to identify disease signatures and drug targets and subpopulations of responders and non-responders to drug treatment [22,89]. Recently, another interesting study used Boolean models to integrate patent data to derive a personalized model, which can reproduce and analyze the gain-of-function or loss-of-function of mutated genes in specific diseases [21]. The model predicted dissimilarity in the PI3K-AKT pathway that resulted in heterogeneity in pancreatic cancer patients. Models also successfully simulated the known dynamics of invasive cancer in the breast and the bladder and predicted disease signatures and possible therapeutic targets that can revert invasive to non-invasive phenotypes [90]. Further, model-predicted signatures were used to stratify patients in long and short survivals.

ODE models have been used to simulate clinical trials and to propose individualized diagnostic and therapeutic targets, such as highly heterogeneous prognostic markers for neuroblastoma and patient stratification into low and high survival based on model simulations [92]. They were also used to predict the patient response to apoptosis-inducing therapeutics and revealed significantly different inter-individual responses [93].

### 3.2. Diagnosis

In diagnosis, the amount of personalized data is frequently sufficient to allow application of ML/DL concepts. In addition, the provided data types are often too heterogeneous, such that a structural presentation of the governing processes is impossible due to an incomplete functional understanding. ML/DL models can make discoveries by analyzing large sets of input data to identify patterns and associations relevant to the outcome of interest. These computational models can function as clinical-decision support systems to assist clinicians in diagnosis and therapy [94,95]. In the following, successful examples for the analysis of patient data with concepts from ML/DL are discussed (summarized in Table 2):

ML has been shown to be an effective analysis tool in a wide variety of medical and biological disciplines. The use of ML models may lead to a better understanding of diseases and the mode of action of drugs as well as to precision medicine. Mason et al. showed that optimized therapeutic antibodies can be found by predicting antigen receptor specificity with ML. It aids in integrating information from several individual receptors into a single pan-receptor model [96]. In metabolomics, Goodwin et al. used Self-Organizing Maps (SOM) to structure microbial metabolic responses, and hence, were able to identify a response metabolome [97].

Thorsen-Meyer et al. showed that a time-sensitive ML model can help predict 90-day mortality in patients administered to the intensive-care unit. In addition, they were able to highlight features contributing to a certain prediction at any time-point, allowing a physician to alter treatment. A further prominent example of ML/DL in clinical application is image analysis [98,99]. ML image analysis is most commonly used for radiological assessment and related decision making in the clinic, and models used can be categorized as either classification tasks or regression models. In the most commonly used classification tasks, models make decisions about categorical end-points such as a disease score or whether a clinician should take a closer look at an X-ray scan. In regression models, the aim is to predict a continuous variable, for example, the survival time of a patient, which often is not categorized into distinct portions. In 2018, Abramoff et al. debuted the first FDA-autonomous AI diagnostic system in medicine [100]. This development marked a milestone as ML models only assisted physicians in their diagnostic decisions previously. Abramoff et al. developed a convolutional neural network (CNN) that was able to correctly classify diabetic retinopathy with a sensitivity of 87.2% and a specificity of 90.7%. In recent months, ML has been widely used to diagnose and detect COVID-19-induced lung pneumonia (i.e., novel COVID pneumonia (NCP)), in computed tomography (CT) images. Zhang et al. developed a multi-scale approach that can not only distinguish NCP from other forms of lung lesions but also predict the progression of a patient into critical illness [101]. They were also able to correlate the features detected in segmented lung scans to clinical parameters like serum C-reactive protein (CRP) and albumin levels.

Another important example for the use of ML models in clinical application is polygenic risk scores (PRS) for common diseases based on findings from genome-wide association studies (GWAS). PRS are estimated to predict the risk of an individual for having a disease based on the individual’s genetic make-up. GWAS that scan the genomes of thousands of individuals offer a very powerful method to identify the multiple genetic risk factors for having the disease that are used in PRS estimation. The potential of PRS is that it can aid in disease prevention and early diagnosis as well as precision medicine, leading to better health outcomes for individuals. There are currently different approaches developed for the estimation of PRS that may lead to some improvement over the other [102,103,104,105,106,107,108,109,110,111,112]. Examples of PRS studies include those on educational attainment [113], schizophrenia [114,115], diabetes [116,117,118], blood pressure [119], depression [120], coronary artery disease, atrial fibrillation, type 2 diabetes, inflammatory bowel disease, and breast cancer [102,116]. Table 3 provides an overview of the discussed modelling approaches and their application in clinical diagnosis.

Further technical details about different ML/DL methodological approaches commonly used (such as convolutional neural network (CNN), random forest (RF), support vector machines (SVM), and K-nearest neighbors (KNN)) and current best-practice recommendations, especially for biological applications, were reviewed and discussed elsewhere in detail [74]. The Polygenic Risk Score Task Force of the International Common Disease Alliance also published a recent comprehensive perspective on PRS with regards to the current application of PRS and challenges faced for improved use in clinical practice [132].

### 3.3. Therapy

Applications in therapy are rather mechanism-based since a functional understanding in the resulting computational simulations is mandatory for exhaustive risk assessment, e.g., in clinical-trial simulations. Moreover, extrapolation between different patient cohorts, treatment schedules, or even species is a frequent requirement in pharmaceutical development. Illustrative examples for applications of mechanistic modeling in therapy are discussed in the following section and summarized in Table 4.

PBPK models can be used to support pediatric investigation plans [28], where PK profiles in children, toddlers, or neonates are simulated based on a reference PBPK model for average grown-ups. The wealth of such pediatric extrapolations lies in the fact that children are not just small adults but differ amongst others in terms of their body composition (fraction of water, fat, and protein, respectively) and a resulting change in drug distribution within various tissues. Similarly, maturation of absorption, distribution, metabolism, and excretion (ADME) of proteins has a significant effect on drug PK in different age groups. Another prominent example for specifications of PBPK models are patient cohorts with hepatic or renal impairment [136,137]. PBPK models for such patient subgroups can be used to simulate and analyze the effect of reduced drug clearance and to assess the resulting increase in drug exposure, which may lead to adverse events. For cirrhotic patients, such concepts have been used to represent pathogenesis according to the Child-Pugh score A to C in terms of, for example, ADME gene expression, plasma protein concentration, or glomerular filtration rate (GFR) rates. Pediatric extrapolations or patients with impaired drug clearance are not fully-personalized models, but they represent an important conceptual application for knowledge-based individualization of mechanistic computational models.

Another concept in pharmacology that is even closer to personalized medicine is model-informed precision dosing (MIPD). MIPD is currently only applied to a few drugs that are subject to therapeutic drug monitoring and for which appropriate PK models are available. Of note, such models basically require the creation of virtual twins, which are individualized based upon information well beyond population demographics [133]. In a recent study, the benefit of using individual patient information to improve PK predictions was systematically assessed [138]. A PBPK model of caffeine was stepwise personalized by using individual data on (i) patient biometry, (ii) patient physiology, and (iii) *CYP1A2* phenotype of 48 healthy volunteers participating in a single-dose clinical study. Model performance was benchmarked against a caffeine base model simulated with parameters of an average individual. The key outcome of the analysis was that consideration of subject-specific data in personalized PBPK models may increase the accuracy of the PBPK predictions but that the degree of improvement is largely dependent on the match between the patient data and the specific pharmacology of the drug.

## 4. Challenges and Recommendations

As outlined above, considerable results have already been achieved with computational models in personalized medicine. However, there are still significant challenges until a full adoption of model-based workflows in clinical research and practice will be reached. Interestingly, despite obvious technical differences between the computational concepts in mechanistic modelling and ML/DL, challenges regarding data availability/harmonization, model development/validation as well as standardization, model re-use and reporting are common to both fields.

### 4.1. Challenges

#### 4.1.1. Data Availability and Data Harmonization

The first step in generating a model is collating the data that need to be integrated into the model. This task heavily relies on the data being formatted and annotated correctly. Reporting and annotation checklists, or “minimum information guidelines” are available for many different kinds of datatypes and can be found and accessed via the FAIRsharing portal (www.fairsharing.org). For availability and data harmonization, both mechanistic modelling and ML/DL fundamentally rely on clinical measurements for both model development as well as independent model validation. A significant challenge for data availability is confused reporting, which makes it difficult to harvest the full benefit of results, navigate the biomedical literature, and generate clinically actionable findings (Varga et al. 2020, under review). As a putative consequence, incomplete sets of data may arise such that specific features have to be left out of the analysis or missing data have to be imputed. Another challenge, particularly in multicentric studies, is the heterogeneity of input data from different laboratories, which, as such, significantly hampers the comparability of results as well as subsequent analyses. Metastandards, such as the novel ISO 20691 “Biotechnology—Requirements for data formatting and description in the life sciences” (https://www.iso.org/standard/68848.html), help in guiding through the consistent formatting and annotation of data used for modelling.

#### 4.1.2. Model Development and Model Validation

A common challenge for model development and model validation is that all computational models are highly context specific; therefore, they cannot be generalized for different scenarios due to limited extrapolability. Common to all in silico models is a need for validation [139] and predictive accuracy. However, model-validation methods are considered to be individual and type-specific. It is important that any algorithm performs well on novel data that have not been used in training the algorithm; i.e., the model should be able to generalize to new data from the same domain [140]. Uncertainty in model parameters as well as inter-individual variability are hard to assess as such significantly impeded model identification. Likewise, there are errors in basic model structure from prior assumptions, group associations, or pre-determined correlations in clinical relationships, which may bias outcomes. For many types of models and corresponding formats, the model setup and simulation environments can be documented in a standardized format, the Simulation Experiment Description Markup Language (SED-ML) that was designed to record such descriptive information necessary to re-run a model, such that it can be exported from one simulation tool and imported into another.

#### 4.1.3. Model Standardization, Model Re-use, and Reporting of Results

Model standardization, model re-use, and reporting of results are also largely hampered by the context specificity of most models. Moreover, many software tools still do not provide open-access rights, long-term maintenance, versioning, or standard software qualification. Likewise, very frequently, best practices are often not applied routinely, even if they are available and supported by several tools, such as the COMBINE archive for simulations [141]. In consequence, there is a lack of practice for standardization model development and the use of community defined formatting standards including Extensible Markup Language (XML)-based machine-readable formats for the model itself. Notable exceptions are SBML as a standardized interchange format for computer models of biological processes; CellML, a standard format to store and exchange reusable, modular computer-based mathematical models; or NeuroML (Neural Open Markup Language), which allows standardization of model descriptions in computational neuroscience and others [142].

#### 4.1.4. Legal and Ethical Issues

Legal and ethical issues include data protection and the anonymity of patient data. Additionally, questions of liability may arise in the case of malfunction, which may be hard to address, and are particularly a constant matter of debate [143]. Legal aspects are largely covered by the general data protection regulation (GDPR). Challenges for computational modelling arise, for example, in data minimization, especially in data-driven models where it is not known which data will be necessary. Likewise, proprietary aspects of data and models may be of relevance. Finally, it may be important to exclude that important decisions are made by automated processing including the right to transparency.

In essence, computational models must go through a process of peer review and validation, and if they are to be adopted by clinicians, they must be assessed through research and state procedures including randomized clinical trials (RCTs) and medical devices regulation (MDR). Source input data should fulfill standard requirements in terms of data quality and representativeness, i.e., for example, inclusion of people of different genders or ethnicities. The generation of large amounts of knowledge about patients beyond the issue patients seek treatment for may be problematic, in particular since this explicitly implies the “right not to know.” Additionally, the success rates for translatability into clinical practice should be openly communicated to the patients.

### 4.2. Recommendations

The four key challenges discussed above can be used to formulate a set of recommendations that apply to different stages of a research project, starting from early ideation to implementation in clinical practice. The recommendations will be discussed in the following (illustrated in Figure 2). Inevitably, these recommendations may remain a little generic for specific studies, given already the complementary character of mechanistic models on the one hand and data-driven approaches on the other. However, the reader is encouraged to compare the recommendations given here in the face of the various examples discussed above (Table 2, Table 3 and Table 4). Additionally, some of the software packages discussed provide best-practice guidelines for study design as well as online resources (Table 2), which can be used for their own applications.

#### 4.2.1. Study Design

At the start of a research project, the study design itself should be defined and agreed on. Additionally, it should be decided whether the to-be-expected amount of data on the one hand and the structural knowledge of the underlying relationships on the other indicates the use of either ML/DL or mechanistic modelling. Ideally, an analysis plan should be formulated to address specific requirements and, above all, to meet the clinical needs of a study. Such preparatory discussions about the study design also support early involvement of all stakeholders and, in particular, their expertise in an interdisciplinary project team. This early-project stage also involves the definition of clear rules about general data protection regulation (GDPR) compliance and data ownership. Additionally, it should be agreed whether the automated processing of data by systems including models is allowed or ruled out such that the physician makes the final decision.

#### 4.2.2. Data Acquisition and Operation

To increase the diversity of data as well as the sample size, multicentric studies are clearly preferable. Additionally, data harmonization needs to be discussed and addressed at the beginning of a study. Informed consent of patients should be taken care of as well. It should be communicated that data donation for the good of future patients and that consent to treatment implies consent to one’s data being processed. Acquisition of patient data also should include structured and standardized reporting of all relevant clinical characteristics measured as well as patient anthropometry, physiology, disease state, and other phenotypic information and further influencing factors. This disease and phenotype information should be shared, respecting data privacy where applicable, and following a standardized structuring format, such as Phenopackets (http://phenopackets.org), linking phenotype descriptions with disease, patient, and genetic information, enabling clinicians, biologists, and disease and drug researchers to build more complete models of disease. If possible, electronic health records of patients should be used, for seamless data interoperability ideally formatted in accepted standard formats, such as Fast Healthcare Interoperability Resources^®^ (FHIR^®^; http://hl7.org/fhir/) of the health-care standards organization Health Level Seven International (HL7). Data should be as comprehensive and unbiased as feasible, rather than using selected data based on potentially flawed existing knowledge. Additionally, data should be available in both processed and unprocessed forms. Data used in creating models and in applying them must be representative, comparable, fairly sourced, and logically analyzed. Representativeness and completeness of data point towards state collection of standardized, harmonized data across the population, provided as a non-commercial resource to research projects, which is the current system in many countries and is provided for by exceptions to the requirement for consent in specific national and EU legislation. Data input into models and model validation should be quality-checked thoroughly and on a running basis, with a high level of transparency, to achieve the best possible level of correct results for patients individually and collectively, as is the case with other tools used by clinicians.

#### 4.2.3. Model Development and Model Validation

For model development and validation, one should start with the simplest model structure, and if results are not sufficient, increase complexity in small steps. A key question during model building always is whether the model generalizes well enough and how it responds to unknown data. This follows the usual concept of training and validation. In general, simulation outcomes should be compared to standard measures for benchmarking of the computational predictions. To support the re-use of results, software tools applied should support documentation as well as versioning. Reporting of simulation results should follow best-practice guidelines. With regard to safety and transparency, data used in healthcare models should include information about the source of data and inspection of data quality, to be ensured by rigorous scientific testing and licensing through well-funded licensing agencies.

Validation of models to be used clinically must include extensive and frequent testing by scientific peers and licensing agencies, as well as continuous quality reporting by centres using these systems. Commercial and scientific application of models involves different routes to validation and are difficult to discuss together, but the testing and validation process should in either case be rigorous, planned, ongoing, and well-financed. Licensing fees can, for example, cover the cost on continual testing of commercial applications. Companies should be required to disclose both data and models to regulatory agencies and/or peer reviewers and journals for the purposes of proper quality control. Third-party validation and replication of results requires access to the data used for the development and training of the models. While access to health-care data used is theoretically possible by application to relevant authorities, preprocessing and harmonizing the data to replicate results is time-consuming and unrewarded. Therefore consistent, high-quality model validation requires that systems be created allowing the return of enhanced (pre-processed, harmonized) medical data to the state.

There are guidelines and methods for validating models, which are accurate and confident in predictions, both in terms of accuracy and confidence in predictions. Performance evaluation of in silico models should be transparent and consistent to existing guidelines, explaining the reasons for not doing so when alternative methods are chosen. Mechanistic models can serve multiple purposes, which makes it difficult to have a strict set of guidelines for validations. Further, for clinical practice, models are still striving to become a reliable partner to understand mechanisms underlying diseases, identify biomarkers (diagnosis and prognosis), and support clinical-treatment decisions. Towards these purposes, the model will pass through a large set of testing and validation procedures to ensure that model predictions hint to physiologically plausible states for realistic levels of parametric uncertainty. After parameter estimation, one has to compare the overall behavior or stimulus-response pattern and match this with evidence from the literature to ensure that the model represents the biological reality. This “matching” is a judgement and a discussion that continues an iterative process of model refinement. Ultimately, the usefulness of a model is not only measured by the accuracy of representation but how well it supports the generation, testing. and refinement of hypotheses and make predictions based on which patients are stratified between responder and non-responder to a specific therapy [144]. Model validation is crucial, and the accuracy of ML/DL results must be verified using independent test datasets. Supervised ML/DL clinical-decision support tools use ML algorithms to make individual patient-level predictions.

#### 4.2.4. Translations and Applications

The final step is translation and application such that the results can actually be brought back to the clinic. Here, the model results may be used for the generation of testable hypotheses or to simulate clinical studies. Likewise, clinical scores and markers may be an outcome. In this regard, it is of utmost importance that the confidence in the computation predictions is frankly disclosed. Likewise, underlying model assumptions and uncertainties should be openly discussed in particular with regard to potential consequences for the patient. This also involves algorithmic fairness. Biases should be addressed through analysis of input data both at the licensing and at the application of the systems to new populations, as indeed, should be the case in all tools used in medicine. We have seen many cases of medical discoveries and tools being used on populations for whom they were not designed, to the detriment of these patients’ treatment and outcomes. Continuous analysis of the applicability of a model to a target population should be a part of data integration and model validation—however, the perfect should not be the enemy of the good, and where data on a given population is missing, the tool should still be used if it can be shown to generate better outcomes for patients than not using it. The GDPR has gone a long way to ensuring that people and corporations dealing with personal data protect them, and these efforts should be continued and redoubled to ensure technical security accompanies legal measures. Technical measures also include data federation and the creation of synthetic data, which can be shared with fewer restrictions. However, real data that have been identified, encrypted, harmonized, standardized, validated, and properly curated are more likely to lead to both better science and patient treatment, and they also offer significantly better transparency and quality control. This should therefore be the gold standard required for models used in treatment, while synthetic data can be used as sandboxes for hypothesis generation.

## 5. Conclusions

There is a rapidly growing amount of personalized data in today’s medicine. The availability of these measurements holds immense promises for both diagnosis and treatment of diseases at the single-patient level. The complexity of the data, however, poses significant challenges in its general usability, amongst others due the underlying heterogeneity of samples as well as inter-patient variability. Computational models provide a structural framework to analyze these data through their contextualization in mathematical descriptions, be it mechanistic modelling or ML/DL. As discussed here, there are numerous successful examples for the application of computational models in discovery, diagnosis, and therapy (Table 2, Table 3 and Table 4). However, several challenges remain to fully realize the possibilities of personalized data in clinical practice, in particular regarding data provision, model building, and model filing as well as legal issues and ethics. To support successful study outcomes, key guidelines can be formulated that refer to different stages of project work:Careful planning of study design is of utmost importance at the project start;Common standards for data sampling, data acquisition, and data operation should be fulfilled;Data harmonization is crucial to ensure data compatibility and comparability;Data should be divided in data sets for training and validation;Model documentation should be written according to best practice guidelines;It is important to openly communicate model assumptions and biases in the computational results;New patient data should be continuously used for benchmarking of the computational results.

We strongly believe that compliance to these guidelines will significantly support the realization of computational precision medicine in the future.

## Figures and Tables

**Figure 1 jpm-12-00166-f001:**
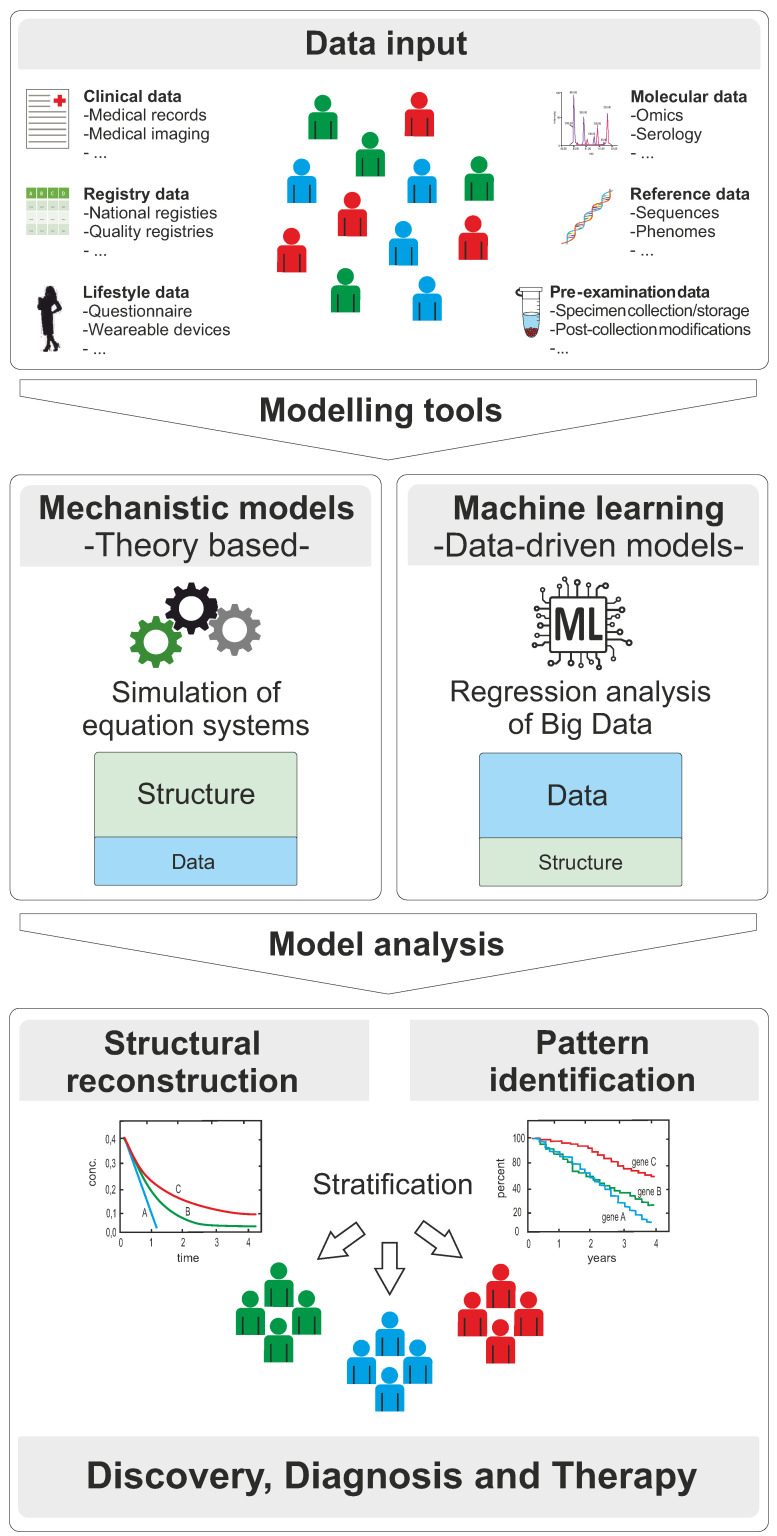
Computational concepts for patient stratification in personalized medicine. The modelling process starts with collection of data from various sources (data input). The two basic modelling tools are depicted as mechanistic models (theory-based) and machine learning (data-driven). Model analysis leads to either a structural reconstruction of physiological mechanisms that drive disease or to pattern identification from large data sets. The information obtained by these approaches can be used to generate knowledge for stratification of patients into specific subgroups facilitating discovery, diagnosis, and therapy in personalized medicine.

**Figure 2 jpm-12-00166-f002:**
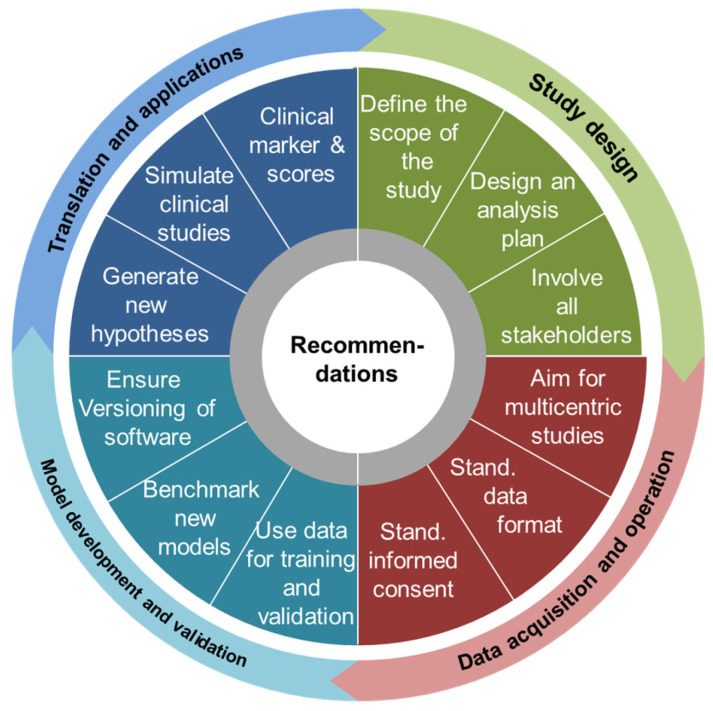
Basic recommendations for the use of computational models from early ideation to implementation in clinical practice. For each of the four key challenges (outer circle), a specific set of basic recommendations is given in the corresponding color. Stand.: Standardized.

**Table 1 jpm-12-00166-t001:** Resources and tools used to construct MIMs, quantitative and qualitative models, and pharmacokinetic models.

Research Field	Resources	Tools
Molecular interaction maps	SIGnaling Network Open Resource (SIGNOR) [40], Reactome [41], SignaLink [42], InnateDB [43], Atlas of Cancer Signalling Network (ACSN) [15], OmniPath [35], RING [36], WikiPathways [44], Kyoto Encyclopedia of Genes and Genomes (KEGG) [45]	CellDesigner [46], Cytoscape & plugins [47], Molecular Interaction NEtwoRks VisuAlization (MINERVA) [48], NaviCell [49], Newt [50]
Boolean models	CellNetAnalyzer (CNA) [37], CellCollective [38], GINsim [39], PyBoolNet repository [46], BioModels [47]	CNA [37], Genetic Network Analyzer (GNA) [48], CellCollective [38], GINsim [39], SQUAD-Boolsim [49], BoolNet [18], Markovian Boolean Stochastic Simulator (MaBoSS) [50], CellNOpt [51]
Constrained-based models	BioModels [47], BiGG (Biochemical, Genetic and Genomic knowledge base) [52], Human metabolic atlas [53], Virtual Metabolic Human [54]	COnstraint-based Reconstruction and Analysis (COBRA) toolbox [55], Sybil package [56], COBRApy [57], ModelSEED [58]
Quantitative models	BioModels [59], Java Web Simulation (JWS) [60], Physiome Model Repository [61]	COmplex PAthway Simulator (COPASI) [62], CellDesigner [63], JWS [60]
Pharmacokinetic models	PharmML (Pharmacometrics Markup Language [30], Open Systems Pharmacology	Monolix, SimCyp^TM^, GastroPlus^®^, PK-Sim^®^

Resources: comprises repositories of manually curated causal interactions and published models that can be used for the construction of new models. Tools: includes software programs with user interface to construct, visualize, or dynamically simulate the models.

**Table 2 jpm-12-00166-t002:** Examples for mechanistic modelling in discovery.

Research Field	Content
Molecular interaction maps	
Inflammation	Knowledge-base, disease mechanisms, data interpretation [83]
Neurodegenerative disease	Knowledge-base, disease mechanisms, data interpretation [14]
Cancer	Knowledge-base, disease mechanisms, data interpretation [15]
Rheumatoid Arthritis	Knowledge-base, critical nodes (drug targets) [84]
Asthma	Disease mechanisms [85]
Atherosclerosis	Disease mechanisms, data interpretation, critical nodes (drug targets) [86]
Boolean models	
Cancer	Disease mechanism, patient stratification [17]
Type 2 diabetes	Disease mechanism, patient stratification [87]
Obesity	Disease mechanism, patient stratification [18]
Non-alcoholic fatty liver disease	Disease mechanism, patient stratification [88]
Genome-scale metabolic models	
Cancer	Disease markers, drug targets, patient stratification [22]
Auto-Immune diseases	Target identification, biomarkers, patient stratification [89]
Cancer	Personalized combination therapy [21]
Cancer	Disease signature, drug targets, patient stratification [90]
Cancer	Disease markers, drug targets, patient stratification [22]
Auto-Immune diseases	Target identification, biomarkers, patient stratification [89]
Cancer	Personalized combination therapy [21]

**Table 3 jpm-12-00166-t003:** Examples for the application of machine learning and deep learning algorithms in diagnosis.

Research Field	Content
Deep Learning and Convolutional Neural Network Models	
Ophthalmology	The first FDA-authorized autonomous AI system for the detection of diabetic retinopathy [100]
Radiology	DL based model that is able to detect COVID-19-induced pneumonia on chest X-ray images [101]
Ophthalmology	Two models for quality assurance and diagnosis of diabetic retinopathy on retinal images [121]
Pathology	Assistance to pathologists for improving classification of lung adenocarcinoma patterns by automatically pre-screening and highlighting cancerous regions prior to review [122]
Imaging flow cytometry	Automated image de-blurring of out-of-focus cells in imaging flow cytometry [123]
Ophthalmology	A DL model for the diagnosis of glaucoma based upon images and domain knowledge features [124]
Oncology	Automated detection of oral cancer on hyperspectral images [125]
Deep Learning and Deconvolutional Neural Network Models	
Proteomics	Neural network that is able to predict signal peptides (SP) from amino-acid sequences and distinguish between three groups of prokaryotic SPs [126]
Antibody engineering	Prediction of antigen specificity via DL, which leads to optimized antibody variants for therapeutic purposes [96]
Intensive care	ML analysis of time-series data in intensive care units led to an improvement in the prediction of 90-day mortality [71]
Deep Learning, Machine Learning, Random Forest, and Deconvolutional Neural Network Models	
Psychiatry	A model that detects autism spectrum disorder risk for newborns with up to 95.62% from electronic medical records [127]
Neurology	A study with the aim to differentiate between cognitive normal people and patients with Alzheimer’s disease using various ML/DL techniques on blood metabolite levels [128]
Machine Learning and Polygenic Risk Score Models	
Coronary artery disease	Patients with high genome-wide PRS for coronary artery disease may receive greater clinical benefit from alirocumab treatment in the ODYSSEY OUTCOMES trial [102]
Coronary artery disease, atrial fibrillation, type 2 diabetes, inflammatory bowel disease, and breast cancer	Genome-wide polygenic scores for common diseases identify individuals with risk equivalent to monogenic mutations. Use of PRS to identify individuals at high risk for a given disease to enable enhanced screening or preventive therapies [116]
Machine Learning, Self-Organizing Maps, Random Forest, K-Nearest Neighbors, Support Vector Machines, Self-Operating Maps	
Metabolomics	SOM analysis of response metabolites detected by mass-spectroscopy leads to the identification of similar responses (ML/Self, Organizing Maps (SOM)) [97]
Imaging flow cytometry	An open-source toolbox for the analysis of imaging flow cytometry images (ML/RF) [129]
Radiology	Classification of COVID-19 and non-COVID-19 patients based on features extracted from chest X-ray images (ML/KNN) [130]
Endocrinology	Prediction of diabetes based on several blood values and other patient indices (ML/SVM, RF) [131]
Metabolomics	SOM analysis of response metabolites detected by mass-spectroscopy leads to the identification of similar responses (ML/SOM)) [97]

CNN: Convolutional Neural Network, RF: Random Forest, DNN: Deconvolutional Neural Network, SOM: Self-Operating Maps, SVM: Support Vector Machines, KNN: K-Nearest neighbors, PRS: Polygenic Risk Score.

**Table 4 jpm-12-00166-t004:** Examples for mechanistic modelling in therapy.

Research Field	Content
Mechanistic Models	
Pediatrics	Pediatric extrapolation [28]
Geriatrics	Geriatric extrapolation [33]
MIPD	Prediction of personalized drug exposure [133]
Pharmaco-genomics	Prediction of the incidence rates of myopathy in different genotypes [134]
Disease models	Prediction of drug PK in cirrhotic patients [135]

MIPD: Model-informed precision dosing, PK: Pharmacokinetic.

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
