# Peer review of "Computational Models for Clinical Applications in Personalized Medicine—Guidelines and Recommendations for Data Integration and Model Validation"

_jpm, 2022, doi:10.3390/jpm12020166_

Round 1

Reviewer 1 Report

The review aims to discuss in detail computational models for personalized medicine. It provides useful examples of how computational models are applied in clinical practice, it discusses open challenges and offers guidelines to create standardized and harmonized personalized. The manuscript is beautifully written. I strongly recommend this review for publication; however, there are a few issues that should be addressed:

  • The review is conceived to provide best practice guidelines for modelling application in clinical care. It provides nice examples of how mechanistic models and machine learning models can be applied in discovery, diagnosis and therapy in personalized medicine. However, it lacks practical guidelines for application of similar approaches in novel contexts. In particular…
  • For the extraction of MIMs, what data resources can be accessed to extract novel data? There is a large number of manually curated databases that provides disease maps in standardized formats (e.g. Wikipathways, SIGNOR database, Reactome… see also the disease-map community) that can be tailored and refined to adapt to new context.
  • What tools can be used to build Boolean/quantitative/pharmacokinetics models?
  • What are the scientific communities that work toward the harmonization of approaches in each of these fields? See also the disease map community for MIMs, the COLOMOTO community for mechanistic models etc..
  • To address the above questions, I would suggest authors to provide useful lists of tools/resources or, in alternative, to mention detailed reviews that’s might support the choice of an inexpert user.
  • Similarly, in Table 2, the authors provide examples for the application of ML and DL algorithms in diagnosis, it would be nice if they could provide a brief description and application areas of individual types of algorithms (Random forest, Deconvolutional Neural Networks etc…) or at least provide a link to a detailed review where they have been described.
  • In the ‘challenges and recommendations’ section authors outline challenges toward a full adoption of model-based workflows in clinical research, and provide recommendations to meet these challenges. However, I have found recommendations a bit vague and generic. I would suggest the authors to add more practical advises, i.e. is there any standard to follow for the study design? What are the standards for model development/ validation?

Concerning the paragraph (lines 517-523) ‘Model validation should again include extensive and frequent testing by scientific peers and licensing agencies, as well as continuous quality reporting by centres using these systems. Licensing fees should be raised to cover these, and compulsory insurance should be required to handle harm done when clinical tools fail. Companies should be required to disclose both data and models to regulatory agencies and/or peer reviewers and journals for the purposes of proper quality control. Likewise, systems should be created for the return of enhanced, state-sourced data, to the state.’

 I’ve found the paragraph a bit out of context here, could the authors explain more accurately how this fit to the model validation issue?

Minor revisions:

Line 54: ‘please replace ‘The share’ with ‘They share’

Line 81: ‘please replace ‘Incorporating’ with ‘By incorporating’

Paragraph 2.1 Mechanistic models, references are missing in the second half of the paragraph

Paragraph 2.1.1 Molecular Interaction Maps, references are missing in the last part of the paragraph

Paragraph 2.1.4 Pharmacokinetics Models, references are missing in the last part of the paragraph

Line 311: check ‘, ,’

Line 555: check ‘..’

Author Response

Dear Sir or Madam,

Thank you very much for giving us the opportunity to submit a revised version of our review "Computational models for clinical applications in personalized medicine – guidelines and recommendations for data integration and model validation". We greatly appreciate the careful review our manuscript received and thank you for your insightful comments that have helped to improve our manuscript.

Changes in the manuscript text are shown in red color. Enclosed you will find a point-by-point reply to your review. In closing, we are pleased by the reviews that our manuscript has received and hope that the revisions we made will now make it acceptable for publication. We are looking forward to your response.

Kind regards,

Lars Kuepfer

Response Reviewer 1

The review aims to discuss in detail computational models for personalized medicine. It provides useful examples of how computational models are applied in clinical practice, it discusses open challenges and offers guidelines to create standardized and harmonized personalized. The manuscript is beautifully written. I strongly recommend this review for publication; however, there are a few issues that should be addressed:

The review is conceived to provide best practice guidelines for modelling application in clinical care. It provides nice examples of how mechanistic models and machine learning models can be applied in discovery, diagnosis and therapy in personalized medicine. However, it lacks practical guidelines for application of similar approaches in novel contexts. In particular…

  • For the extraction of MIMs, what data resources can be accessed to extract novel data? There is a large number of manually curated databases that provides disease maps in standardized formats (e.g. Wikipathways, SIGNOR database, Reactome… see also the disease-map community) that can be tailored and refined to adapt to new context.
  • What tools can be used to build Boolean/quantitative/pharmacokinetics models?
  • What are the scientific communities that work toward the harmonization of approaches in each of these fields? See also the disease map community for MIMs, the COLOMOTO community for mechanistic models etc..
  • To address the above questions, I would suggest authors to provide useful lists of tools/resources or, in alternative, to mention detailed reviews that’s might support the choice of an inexpert user.

We are grateful to the reviewer for these suggestion! To address above four points, we have now included an additional section (2.1.5.) on resources and tools (page 5ff). This paragraph also includes an additional table (Table 1).

  • Similarly, in Table 2, the authors provide examples for the application of ML and DL algorithms in diagnosis, it would be nice if they could provide a brief description and application areas of individual types of algorithms (Random forest, Deconvolutional Neural Networks etc…) or at least provide a link to a detailed review where they have been described.

This remark is also very valid! We have included an additional explanation in the Discovery section (3.2) and referenced several instructive reviews:

“Further technical details about different ML/DL methodological approaches commonly used (such as CNN, RF, SVM and KNN) and current best practice recommendations, especially for biological applications, were reviewed and discussed elsewhere in detail [78]. Polygenic Risk Score Task Force of the International Common Disease Alliance also published a recent comprehensive perspective on PRS with regards to the current application of PRS and challenges faced for improved use in clinical practice [125].”

  • In the ‘challenges and recommendations’ section authors outline challenges toward a full adoption of model-based workflows in clinical research, and provide recommendations to meet these challenges. However, I have found recommendations a bit vague and generic. I would suggest the authors to add more practical advises, i.e. is there any standard to follow for the study design? What are the standards for model development/ validation?

The reviewer is certainly right, here. However, given the heterogeneity of the various approaches presented in this review we feel this is rather unavoidable. We now included an additional paragraph to provide some guidance to the reviewer for the design of own studies in the future (page 15, line 421ff).   

“Inevitably, these recommendations may remain a little generic for specific studies, given already the complementary character of mechanistic models on the one hand and data-driven approaches on the other. However, the reader is encouraged to compare the recommendations given here to the various examples discussed above (Tables 2-4) . Also, best practice guidelines for study design are provided with some of the software packages and online resources (Table 2) which can be used for own applications.”   

  • Concerning the paragraph (lines 517-523) ‘Model validation should again include extensive and frequent testing by scientific peers and licensing agencies, as well as continuous quality reporting by centres using these systems. Licensing fees should be raised to cover these, and compulsory insurance should be required to handle harm done when clinical tools fail. Companies should be required to disclose both data and models to regulatory agencies and/or peer reviewers and journals for the purposes of proper quality control. Likewise, systems should be created for the return of enhanced, state-sourced data, to the state.’ I’ve found the paragraph a bit out of context here, could the authors explain more accurately how this fit to the model validation issue?

This paragraph has been revised (p15 lines 459ff). It now reads:

“Validation of models to be used clinically must include extensive and frequent testing by scientific peers and licensing agencies, as well as continuous quality reporting by centres using these systems. Commercial and scientific application of models involves different routes to validation, and are difficult to discuss together, but the testing and validation process should in either case be rigorous, planned, ongoing and well-financed. Licensing fees can for example cover the cost on continual testing of commercial. Companies should be required to disclose both data and models to regulatory agencies and/or peer reviewers and journals for the purposes of proper quality control. Third party validation and replication of results requires access to the data used for development and training of the models. While access to health care data used is theoretically possible by application to relevant authorities, preprocessing and harmonising the data to replicate results is time-consuming and unrewardedTherefore consistent, high-quality model validation requires that systems be created allowing the return of enhanced (pre-processed, harmonised) medical data to the state.”

Minor revisions:

  • Line 54: ‘please replace ‘The share’ with ‘They share’

This was corrected.

  • Line 81: ‘please replace ‘Incorporating’ with ‘By incorporating’

This was changed

  • Paragraph 2.1 Mechanistic models, references are missing in the second half of the paragraph

This was corrected.

  • Paragraph 2.1.1 Molecular Interaction Maps, references are missing in the last part of the paragraph

This was corrected.

  • Paragraph 2.1.4 Pharmacokinetics Models, references are missing in the last part of the paragraph

This was corrected.

  • Line 311: check ‘, ,’

This was corrected.

  • Line 555: check ‘..’

This was corrected.

Reviewer 2 Report

This review summarized several computational models for clinical applications. The authors also discussed challenges in data availability, integration as well as model development. Overall, it is well written, but several issues need to be addressed.

  • The computational models are classified as mechanistic model as well as data-driven model, and are complementary, in general. I would suggest the authors to discuss this point in details. For example, guidelines to choose the specific type of model. They can also demonstrate the complementarity of both models in a specific example.
  • In line 173, the authors mentioned interpretable ML/DL model, this is an important research topic because biologists/clinicians need clear causal understanding behind the prediction, thus it receives more attentions in recent years. I would suggest the authors to discuss this issue in the scope of personalized medicine.
  • Line 141 should be "2.1.5"

Author Response

Dear Sir or Madam,

Thank you very much for giving us the opportunity to submit a revised version of our review "Computational models for clinical applications in personalized medicine – guidelines and recommendations for data integration and model validation". We greatly appreciate the careful review our manuscript received and thank you for your insightful comments that have helped to improve our manuscript.

Changes in the manuscript text are shown in red color. Enclosed you will find a point-by-point reply to your review. In closing, we are pleased by the reviews that our manuscript has received and hope that the revisions we made will now make it acceptable for publication. We are looking forward to your response.

Kind regards,

Lars Kuepfer

Response Reviewer 2

This review summarized several computational models for clinical applications. The authors also discussed challenges in data availability, integration as well as model development. Overall, it is well written, but several issues need to be addressed.

  • The computational models are classified as mechanistic model as well as data-driven model, and are complementary, in general. I would suggest the authors to discuss this point in details. For example, guidelines to choose the specific type of model. They can also demonstrate the complementarity of both models in a specific example.

The comparison between mechanistic and data-driven models is a key aspect of the review.  This is now further elaborated on on page 3, line 59ff:

“In consequence, mechanistic models require a structural understanding of a process, yet the demand for data can be limited. Concepts in machine learning, in turn, are fundamentally based on large data sets yet these models do not necessaarily need any prior functional understanding (Fig. 1).”

  • In line 173, the authors mentioned interpretable ML/DL model, this is an important research topic because biologists/clinicians need clear causal understanding behind the prediction, thus it receives more attentions in recent years. I would suggest the authors to discuss this issue in the scope of personalized medicine.

This is a valid point brought up by the reviewer! We have now included an additional explanation on different clinical aspects of interpretable ML/DL models (p.6, line 169ff).

  • Line 141 should be "2.1.5"

This was corrected. Thank you very much!

Round 2

Reviewer 2 Report

The revision has addressed the concerns and is suitable for publication.